# Antioxidant, Antimicrobial, and Kinetic Studies of Β-Cyclodextrin Crosslinked with Lignin for Drug Delivery

**DOI:** 10.3390/pharmaceutics14112260

**Published:** 2022-10-22

**Authors:** Narcis Anghel, Violeta Melinte, Iuliana Spiridon, Mihaela Pertea

**Affiliations:** 1Petru Poni Institute of Macromolecular Chemistry, 41 A Grigore Ghica Voda Alley, 700487 Iasi, Romania; 2Department of Plastic Surgery and Reconstructive Microsurgery, Grigore T. Popa University of Medicine and Pharmacy of Iasi, St. Spiridon Emergency County Hospital, 700115 Iasi, Romania

**Keywords:** lignin, cyclodextrin, antioxidant, ketoconazole, piroxicam, drug delivery

## Abstract

β-Cyclodextrin was attached to lignin/lignin crosslinked by epichlorohydrin and served as a drug delivery matrix. Ketoconazole and piroxicam were added into the polymeric matrix as antifungal and anti-inflammatory agents, respectively. The percentage of drug retained ranged from 48.4% to 58.4% for ketoconazole and piroxicam, respectively. It was found that their tensile strengths increased with decreasing particle size, ranging between 59% and 71% for lignin crosslinked with β-cyclodextrin base matrix (LCD). Depending on the polymeric matrix, the drug release kinetics fit well in the Korsmeyer–Peppas model, with or without Fickian diffusion. From the materials based on the mixture of epoxidized lignin and β-cyclodextrin, the medicines were released more slowly (the release rate constant presents lower values ranging between 1.117 and 1.783), as compared with those comprising LCD (2.210–4.824). The materials were also demonstrated to have antimicrobial activity. The antioxidant activity of LCD loaded with piroxicam was found to be 23.9% greater than that of the base matrix (LCD). These findings could be useful towards β-cyclodextrin attached to lignin formulation development of drug carriers with antioxidant activity.

## 1. Introduction

Lignin, which represents around thirty percent of the carbon in the biosphere, is one of the most significant biopolymers [1]. After delignification processes, lignin could become a source of valuable materials due to its renewable character, low cost, and physiochemical properties. The global pulp industry produces roughly 70 million tons of lignin annually [2], and this material might be used for more than just burning to produce electricity.

Guaiacol, p-hydroxycinnamoyl alcohol, and syringic alcohol are the three different types of phenylpropane precursors that make up lignin. These three precursors are related to one another mostly via randomized (–O–4) aryl ether linkages and carbon–carbon bonds. Phenolic hydroxyl, alcoholic hydroxyl, and carboxyl groups present in lignin influence its reactivity, solubility, stability, and, finally, the potential application of this complex polymer. The heterogeneity of lignin is a result of the multiple biomass sources and extraction methods and represents a factor that limits its uses. The primary uses of lignin and its derivatives in a variety of industries, including the cement industry, agriculture, nanocomposite, and energy materials, have been described [3], while in recent decades, new potential uses were identified in biomedical, hydrogen source, biosensor, and bioimaging fields [4,5,6]. It is worth mentioning that hydroxyl groups present in lignin confer antioxidant properties comparable to synthetic commercial antioxidants such as guaiacol [7]. Lignin’s antioxidant properties make it suitable for new applications in pharmaceutics, cosmetics, or tissue engineering applications.

Cyclodextrins are cyclic oligosaccharides composed of 6–8 glucopyranose units connected by glycosidic linkages. They resemble truncated cones with a hydrophobic internal cavity. Due to its hydrophobic central cavity, β-cyclodextrin may interact noncovalently with various guest molecules to generate inclusion complexes in aqueous solutions. The most common way to bind β-CD to a substrate is by either its functionalization or the use of certain crosslinking agents [8]. Cyclodextrins are utilized as release-modifying additives because they can increase the physico-chemical stability of drug molecules or minimize negative drug responses, according to certain writers [9]. As an alternative, a different method for generating inclusion complexes has been proposed: supercritical fluid technology (SCF). SCF permits the making of inclusion complexes of sensitive compounds under less aggressive circumstances. Active materials are less likely to degrade as a result [10]. In this context, Franco and De Marco used β-cyclodextrin to create inclusion complexes including nimesulide and ketoprofen, two anti-inflammatory medicines, through SCF [11].

Herein, epichlorohydrin has been used as a crosslinking agent for lignin and β-cyclodextrin was attached to lignin/lignin crosslinked by epichlorohydrin. We are aware of relatively little information regarding the use of lignin as a drug delivery matrix and its crosslinking by epichlorohydrin. One existing study [12] refers to the preparation of a hydrogel based on cellulose crosslinked with epoxidized lignin and demonstrates how it works for incorporating and releasing grape polyphenols in a regulated manner. As stated by Culebras et al. [13], wood-derived hydrogels were created in a similar manner as a platform for drug release systems, in which case the crosslink of lignin with cellulose through epichlorohydrin increased the release of paracetamol compared to pure cellulose hydrogel. Lignin added to the hydrogels decreased the interaction of paracetamol with cellulose and increased the diffusion of paracetamol into the medium. The amount of lignin in the cellulose hydrogel greatly altered the drug release behavior, which demonstrated how the hydrogel composition can affect drug release.

As a result of its hydrophobic core cavity, β-CD can form inclusion complexes in an aqueous solution by noncovalent interaction with a variety of guest molecules. As guest molecules, we have used ketoconazole and piroxicam. It was expected that the insertion of β-CD units would enhance the potential of lignin as carriers of ketoconazole and piroxicam.

These new lignin-based materials were investigated in terms of mechanical, antifungal, and anti-inflammatory properties. After the addition of ketoconazole and piroxicam, in order to fit experimental release drug data, the Korsmeyer–Peppas model was used. 

## 2. Materials and Methods

### 2.1. Materials

LignoBoost lignin (LIG) was separated by kraft cooking of softwood according to Spiridon and Tanase [14]. β-cyclodextrin (β-CD), epichlorohydrin (EPI), ketoconazole (K), and piroxicam (P) were purchased from Sigma Aldrich. All other reagents and solvents were of analytical grade and were used without further purification.

### 2.2. Lignin Modification

Epichlorohydrin was crosslinked with lignin in the manner shown in Figure 1: in a 250 mL three-necked flask with a reflux condenser and a magnetic stirrer, 5 g of lignin, 5 g of NaOH, and 50 mL of distilled water were added. After that, the reaction mixture was dropwise added to 50 mL of EPI, and it was agitated for 3 h at 80 °C. The precipitate underwent filtering, methanol washing to eliminate any remaining EPI, and distilled water. For 24 h, the product was vacuum dried at 40 °C. 

The lignin modified by EPI (Lep) was obtained. The subsequent steps for creating epoxidized lignin linked with CD (LCD) were as follows: 3 g of Lep, 9 g of NaOH, and 100 mL of distilled water were combined and agitated for one hour at room temperature. Then, various concentrations of β-CD were introduced. The reaction mixture was brought to 50 °C and stirred continuously for 6 h. The precipitate that was produced was separated by filtration and cleaned with distilled water. The solid was vacuum-dried at 40 °C for 24 h.

### 2.3. Loading of Drugs into LCD Matrix

Firstly, 300 mg of LCD was stirred for 24 h with 50 mL of distilled water containing 50 mg of each drug. Then, the solid was filtered and dried at 40 °C in a vacuum. The amount of bioactive principle encapsulated in cyclodextrin was determined spectrophotometrically by reading the absorbance of the solution before and after filtration at 285 nm for piroxicam and 254 nm for ketoconazole, respectively. The drug quantities incorporated were 24.2 and 29.2 mg of ketoconazole and piroxicam, respectively.

### 2.4. Obtaining of Materials

Materials were made by blending Lep (225 mg), CD (25 mg), and K/P (25 mg) in order to get capsules (named LepCD-K and LepCD-P) with an 11 mm diameter and 1 mm thickness by using a Carver Hydraulic Laboratory Press Model at a ram pressure of 6 tons for 2 min. In the same way, the tablets containing lignin, such as (LIG), lignin coupled with cyclodextrin (LCD), and lignin coupled with cyclodextrin loaded with piroxicam or ketoconazole (LCD-P, LCD-K), were obtained, in this case using 275 mg of each material. LepCD consisted of 250 mg Lep and 25 mg CD.

### 2.5. Fourier Transform Infrared Spectroscopy (Ftir)

The materials’ FTIR spectra were captured using a Brüker Vertex 70 FTIR spectrometer with an ATR (attenuated total reflectance) device (ZnSe crystal) at a 45-degree angle of incidence. With an average of 64 scans and a spectral resolution of 2 cm^−1^, the spectra in the regions of 4000–400 cm^−1^ and 4500–600 cm^−1^ were examined.

### 2.6. Diametral Tensile Measurements

The specimens were removed from the molds and incubated for 1 day at 37 °C with 100% humidity. Shimadzu, Kyoto, Japan’s EZ-Test equipment was used for the diametral tensile strength (DTS) testing, with a loading rate of 0.5 mm per minute. Equation (1) was used to obtain the DTS value:(1)$$DTS=\frac{2\times P}{\pi \times D\times T}$$
where *P* represents the peak load (Newtons), D represents the specimen’s diameter (in mm), and T represents its thickness (in mm). The collected load-deflection curves were used to determine the maximum compression load at failure. The value that was provided was the mean of five measurements.

### 2.7. Dynamic Vapor Sorption (Dvs)

An IGAsorp device (Hidden Analytical, Warrington, UK) was used to assess the water sorption at atmospheric pressure. The investigations had an accuracy of 1% for 0–90% RH and 2% for 90–95% RH, and they were conducted at humidity levels between 0 and 95%, in a temperature range between 5 °C and 85 °C.

The specific surface area was determined using the Brunauer–Emmett–Teller (BET) Equation (2) method based on isothermal investigations [15]:(2)$$W=\frac{W_{m}\times C\times \frac{p}{p_{0}}}{\left(1-\frac{p}{p_{0}}\right)\times \left(1-\frac{p}{p_{0}}+C\times \frac{p}{p_{0}}\right)}$$
where W is the amount of water adsorbed, $W_{m}$. is the amount of water adsorbed in a monolayer, C is the BET constant, and *p* and *p_0_* are the equilibrium and saturation pressure of adsorbates at the temperature of adsorption.

### 2.8. In Vitro Release Studies

Release tests were conducted at a pH of 7.4 and a temperature of 37 ± 0.5 °C. A Jenway 6405 UV-Vis spectrophotometer was used to assess aliquots of the medium that were taken out at regular intervals and were analyzed at a λ_max_ value of 285 nm and 254 nm for piroxicam and ketoconazole, respectively. The kinetics of the drug release were examined.

### 2.9. Anti-Inflammatory Activity

Using a modified version of the protein denaturation (bovine albumin) method reported by Gunathilake et al. [16], the anti-inflammatory activity of the materials was calculated. The reaction mixture was made up of 100 mg of each of the developed materials (LepCD-P and LCD-P) in 5 mL of saline phosphate buffer (PBS, pH = 6.4) and 2 mL of a 0.1% solution of bovine albumin. The mixture was heated to 70 °C for 5 min after being incubated at 37 °C in a water bath for 15 min. The absorbance was measured at 660 nm using a PBS solution as a blank after cooling. The control was a solution of bovine albumin. Each experiment was carried out in triplicate, and Equation (3) was used to calculate the anti-inflammatory activity as percentage inhibition.
(3)$$\%\;inhibition=100\times \left(1-\frac{A_{s}}{A_{c}}\right)$$
where $A_{s}$ is the absorption of the sample and $A_{c}$ is the absorption of the control. 

### 2.10. Antimicrobial Activity

Using the strains of *Staphylococcus aureus* (ATCC 25923), *Escherichia coli* (ATCC 25922), and *Candida albicans* (ATCC 90028), the antimicrobial activity of the materials was evaluated. 

Microbial strains were used to make suspensions in peptone saline solution with a turbidity of 1° McFarland. By dilution, a suspension of 1500 UFC (colony-forming units/mL) was produced. A volume of 10 µL inoculum of the test strains was applied to the surfaces of the testing materials and the control sample. After being retrieved using a sterile swab soaked in peptone saline, the inoculum was sown on the surface of the particular medium. The injected plates underwent a 24 h incubation period at 37 ± 1 °C. The colonies were counted and contrasted with the control sample.

### 2.11. Antioxidant Activity (DPPH Assay)

The DPPH assay was carried out in accordance with the procedure described by Jan et al. [17]. Firstly, 100 mL of methanol was used to dissolve 2.4 mg of DPPH. Samples of each substance weighing 150 mg each were dissolved in a solution made up of 1 mL of the aforementioned mixture and 25 mL of distilled water. After 60 min of dark incubation, the mixture was tested for absorbance at 517 nm. The experiment was run three times, and the standard deviation was calculated. Equation (4) was used to determine the DPPH scavenging activity of different fractions.
(4)$$\%inhibition=\left(\frac{A_{control}-A_{sample}}{A_{control}}\right)\times 100\;$$
where *A_control_* and *A_sample_* were the absorption intensities for blank and sample probes.

## 3. Results and Discussions 

### 3.1. FTIR Analysis of Materials

Figure 1 shows the FTIR spectra of lignin (a), Lep (b), and LCD (c). The FTIR spectrum of lignin shows peaks associated with O–H stretching vibrations (3440 cm^−1^); C–H stretching of –CH_3_; –CH_2_– groups (2950–2835 cm^−1^); C=O stretching vibrations of ketone group (1700–1560 cm^−1^); and peaks at 1590, 1514, and 1410 cm^−1^ associated with the stretching vibrations of the C–C bonds in the aromatic skeleton. Peaks at 1375, 1265, and 1220 cm^−1^ caused by the stretching vibrations of C–O are also noted. These outcomes are consistent with the data that have been published [18]. 

The signal at 760 cm^−1^ caused by the epoxy ring of Lep has vanished (Figure 1c), showing the complete consumption of the epoxy groups. The band at 853–864 cm^−1^ caused by C–H in plane deformations in aromatic rings is seen for both LCD and lignin, demonstrating the samples’ aromatic character. Therefore, it can be stated from the FTIR spectra that the grafting of β-CD onto the surface of lignin was successful.

The condensation index of each lignin sample was evaluated according to Zhang et al. [19] and computed based on FTIR data using Equation (5):(5)$$Condensation\;index=\frac{Sum\;of\;all\;minima\;between\;1500\;and\;1050\;{\mathrm{cm}}^{-1}}{Sum\;of\;all\;maxima\;between\;1600\;and\;1030\;{\mathrm{cm}}^{-1}}.\;$$

The computed values for the condensation index are summarized in Table 1.

LCD shows a degree of condensation approximately 16.2% higher than LIG because the same CD molecule can react with several oxiranic groups.

### 3.2. Mechanical Properties

As seen in Figure 2, the presence of piroxicam and ketoconazole in the lignin-β-CD matrix revealed an increase in DTS (diametral tensile strength), which can be explained by the decrease in the pore size. Similar results were obtained when these drugs were added to xanthan-alginate [20]. Lignin crosslinking by epichlorohydrin means longer polymer chains that tend to decrease free volume and, as a consequence, present limited mobility. Thus, DTS recorded an increment of 136% for material comprising lignin crosslinked by epichlorohydrin due to higher intermolecular stiffness.

The addition of drugs in lignin crosslinked with the β-CD matrix kept the same trend, while DTS slowly increased. The highest DTS was recorded for LepCD-P. As other authors reported [21,22], there is a correlation between mechanical properties and specific surface areas of the powders used to obtain compact materials.

### 3.3. Dynamic Vapor Sorption (DVS)

Sorption water studies were carried out in order to investigate how the sorption capacity is influenced by the components of materials. The interaction of matrix polymers with water is influenced by the availability of polar groups, which can decrease interactions with the added drugs.

Our data evidenced that presence of a guest particle in LCD formulations reduced the water sorption capacity. The observed difference between incorporating K and P can be attributed to differences in the surface area of K as compared to that of P. It seems that more molecular interactions between the polymeric matrix and P as compared with those established with K were present, which reduced the number of available polar functional groups for interactions with water. These interactions affect the molecular mobility of the system and were confirmed also by the evolution of tensile strength.

The average pore size decreased for all materials when drugs were added to the matrix. This may be an explanation for the higher specific surface area of materials available for water sorption.

The lignin crosslinked by epichlorohydrin influenced the sorption capacity of materials. Thus, the material comprising LepCD presented an increase of 12.9% in higher sorption capacity. When ketoconazole was added, the sorption capacity decreased by 5.8%, while the addition of piroxicam resulted in a decrease of 31.8% of this parameter.

According to Table 2 and Figure 3, the addition of ketoconazole in the lignin-β-CD matrix induced a significant decrease in average pore size, as well as an increment in the BET area of LCD-K. It is possible that the binding sites available for water were reduced when K was added (hydrogen bond donor/acceptor: K–0/6 and P–2/6). The same trend was recorded for LCD-P material, but the differences were less significant and must have been due the amorphous nature of ketoconazole [23] and the semi-crystalline nature of piroxicam [24]. The establishment of new hydrogen bonds between the polymer matrix and drugs thus reduced the water adsorption capacity.

Based on the behavior in the water vapor sorption, it can be summarized that materials based on lignin crosslinked by epichlorohydrin-β-CD matrix have a more hydrophobic surface as compared to materials based on the lignin-β-CD matrix. Given that the composition is designed for continuous release, any alterations brought on by moisture uptake are of special significance.

### 3.4. In Vitro Release Studies

We also investigated how medicines were released from the polymeric matrix. It is common knowledge that mathematical models play a crucial role in analyzing medication release mechanisms. The purpose of kinetics models is to specify the release process and make it possible to measure some important parameters, such as the exponent of release.

For all examined materials, the experimental data were best suited by the Korsmeyer–Peppas model, which was created for drug release from polymeric systems specifically regulated by the diffusion mechanism (Table 3, Figure 4). The Korsmeyer–Peppas model was conceived to characterize the drug release from polymeric structures using Equation (6).
(6)$$\frac{M_{t}}{M_{\infty }}=k\times t^{n}$$

The amount of the medication in the initial state is M, the amount released at time t is denoted by $M_{t}$, the release rate constant is k and is represented in min^−n^, and the exponent of release as a function of time t is denoted by n (dimensionless).

Drug release is controlled by diffusion when the exponent of release is equal to 0.5, but when it is between 0.5 and 1, it implies non-Fickian diffusion (drug release is governed by the swelling or relaxation of the polymeric chain).

The LCD-based materials have larger values for the release rate constant (between 2.210 and 4.824) than the Lep-based materials do (between 1.117 and 1.789). Lep and β-CD mixture lower the pace at which active principles are released through hydrophilic interactions (the interactions between –OH groups and the hydrophilic parts of the drugs). Piroxicam and ketoconazole are, thus, released from the material based on LCD, where medicines are entrapped into the β-CD cavity and retained there by hydrophobic interactions, at a little faster pace than from that made up of Lep and β-CD. Due to its minor hydrophily, LCD-P releases piroxicam more quickly than the β-CD cavity because it is hydrophobic. P is released in 600 min in roughly the same quantity.

Ketoconazole was better absorbed by the LCD polymer matrix, as expected, given that it is hydrophobic. Values of the parameter n lower than 0.5 for the LCD matrix show that drug release is controlled by diffusion and not due to swelling and relaxation of the matrix, as in the case of LepCD.

### 3.5. Anti-Inflammatory Activity

Inflammation is a tissue’s response to noxious stimuli, including pathogens. According to data presented in Table 4, the anti-inflammatory activity of LepCD-P decreased by 37.5% as compared with that of LCD-P. This suggests that lignin, a natural polymer separated from biomass [26], and piroxicam could inhibit various pro-inflammatory cytokines. Higher percentage inhibition of LCD-P could be correlated with its higher anti-inflammatory efficacy, having in mind that oxidative damage to biomolecules triggers the inflammation process. The results are consistent with the release studies, with piroxicam being more strongly retained by the LepCD matrix due to the formation of hydrogen bonds with –OH or oxiranic groups. This led to a lower anti-inflammatory activity obtained in the same time interval with LCD-based materials.

### 3.6. Antimicrobial Activity

It is known that phenolic substances in lignin damage the cell membrane at contact with bacteria [27]. Bacteria died as a result of cell membrane rupture and release of cell contents. As result, the bacteriostatic effect is improved. *Staphylococcus aureus* and *Listeria monocytogenes* were two gram-positive bacteria that lignin from maize stover was shown to have distinct antibacterial action against by Dong et al. [28], but not gram-negative bacteria (*E. coli* O157:H7 and *S. Enteritidis*). The antibacterial characteristics of lignin polyurethane/Ag composite foam were reported by other authors to be effective against *Escherichia coli* within 1 h and *Staphylococcus aureus* within 4 h [29]. 

Some authors [30,31] stipulated that groups containing oxygen in lignin impact, and presence of phenolic compounds improve, the antibacterial performance by affecting its antioxidant activity. Therefore, a lower antimicrobial activity for material comprising lignin crosslinked by epichlorohydrin was expected. However, Rocca et al. [32] observed that the layer of bacterial cell walls consisting of peptidoglycan can interact with sugar molecules, which suggests that the sugar content of the lignin material may enhance the adherence to the bacterial membrane. This increased the antimicrobial activity. Our results evidenced a high antimicrobial activity for all materials. LepCD-K presented the highest activity against all studied pathogenic agents.

Gram-negative bacteria were more resistant to our materials than Gram-positive bacteria, as seen in Figure 5. *E. coli* may have displayed more resilience because of its outer membrane [33]. 

### 3.7. Antioxidant Activity

The ability of the studied materials to scavenge free radicals was used to evaluate their antioxidant activity. Usually, this is mainly due to the phenolic hydroxyls present in the lignin structure, which can neutralize free radicals by electron or proton transfer [34]. Our results revealed 42.06% inhibition for the lignin sample (Table 5). 

Depending on the concentration of lignin, several publications [35] found an improvement over radical reduction. Indeed, when lignin was crosslinked by epichlorohydrin, its ability to scavenge free radicals decreased by 37.39%, proving that the decrease of phenolic hydroxyls and the variety of functional groups from the lignin structure have influenced the scavenger activity [36].

When lignin crosslinked by epichlorohydrin was coupled with β-CD, the inhibition degree was recorded at 11.82%. This parameter highly increased to 55.39% when K was added, due to the antioxidant character of ketoconazole [37,38].

## 4. Conclusions

New materials consisting of β-Cyclodextrin attached to lignin/lignin crosslinked by epichlorohydrin as a matrix were used as a drug delivery system for ketoconazole and piroxicam. It was found that their tensile strength increased with decreasing particle size, ranging between 59% and 71% for LCD base matrix, and 8.5% and 38.5% for LepCD. More molecular interactions between the polymeric matrix and P as compared with those established with K were present and, as result, water sorption capacity was lower while tensile properties were higher for materials comprising P. The Korsmeyer–Peppas model, which accounts for diffusion depending on the nature of the polymeric matrix, was well-fitted by the drug release kinetics. LepCD-based materials released the pharmaceuticals more slowly (*k*=1.117–1.789) than LCD-based ones (the release rate constant between 2.210 and 4.824). The materials exhibited antimicrobial activity, also. The antioxidant activity of LCD-P was found to be 23.9% greater than these of the base matrix (LCD). These findings could be useful towards β-Cyclodextrin attached to lignin formulation development of drug carriers with antioxidant activity.

## Data Availability

Data contained within the article are available from authors upon request.

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
