# Peer review of "Antioxidant, Antimicrobial, and Kinetic Studies of Β-Cyclodextrin Crosslinked with Lignin for Drug Delivery"

_pharmaceutics, 2022, doi:10.3390/pharmaceutics14112260_

Round 1
Reviewer 1 Report
In this paper, the authors proposed the crosslinking of beta-cyclodextrin with lignin to obtain a carrier for drug delivery. The paper is interesting but it has to be improved before publication.
Innovative techniques based on the use of supercritical fluids allow the formation of inclusion complexes at micrometric dimensions. They should be named in the introduction (see, for example, 10.1016/j.jcou.2020.101397 and 10.1007/s10847-019-00970-2).
Lines 66-67: The authors wrote that “there are few data related to the lignin 66 crosslinked by epichlorohydrin and its use as matrix for drug delivery”. Those data have to be discussed in the introduction and the related references added.
Figure 1 caption: LIG (1) should be LIG (a).
Figure 4: please add experimental points and fitting after 500 min. It seems to me that the curves are far from the plateau.
Reviewer 2 Report
The authors present some investigations concerning CD crosslinked with lignin in the current study.
In my opinion, the manuscript requires some improvement and the authors should explain some ambiguities before the acceptance of this paper.
First of all the authors provided "non-published material" which contains figures and scheme presented in the manuscript. Is it a mistake?
All structures on the scheme should have their numbers/symbols
How the authors could explain performed anti-inflammatory assay? In the performed experiment the heat-induced denaturation of BSA can be observed. In chapter 3.5 the authors write about noxious stimuli inducing inflammation such as pathogens and oxidative stress but in the performed study they used heat which caused protein denaturation. In my opinion, based on such a method, we could not state the anti-inflammatory activity. . Therefore in my opinion this experiment should be repeated and the authors should use another model such as LPS-induced inflammation.
Why did authors choose the piroxicam which is an old drug with low significance in current medicine and pharmacology?
Why did the authors use anti-fungal ketoconazole in the antimicrobial assay and used it to estimate the activity against bacteria? In my opinion, this is a serious fault. How the authors could justify their choice?
The quality of presentations could be improved and some grammar and spelling mistakes must be corrected
Round 2
Reviewer 1 Report
In my opinion, the paper has been improved and can be accepted as it is
Reviewer 2 Report
The authors have addressed most of my comments.
I still do not understand why figures and scheme are put additionally and separately in the "non-published" file, as they have already been presented in the manuscript.
Moreover, I am not fully convinced by the explanation of the method used for anti-inflammatory evaluation. I advise the authors to use more recognized, unambiguous methods in the future.
Nevertheless, taking into account some improvements in the manuscript I think it can be accepted despite some weaknesses if the Editor deems it appropriate